cognition/psychology

cognition, object representation, language, EEG, infancy

**Author for correspondence:**
Barbara Pomiechowska
e-mail: pomiechowskab@ceu.edu

# Nonverbal category knowledge limits the amount of information encoded in object representations: EEG evidence from 12-month-old infants

## Barbara Pomiechowska[1] and Teodora Gliga[2]

[1]Cognitive Development Center, Department of Cognitive Science, Central European University, Nador utca 9, Budapest 1051, Hungary
[2]School of Psychology, University of East Anglia, Norwich Research Park, Norwich NR4 7TJ, UK

 BP, 0000-0002-3819-7641; TG, 0000-0001-8053-7286

To what extent does language shape how we think about the world? Studies suggest that linguistic symbols expressing conceptual categories ('apple', 'squirrel') make us focus on categorical information (e.g. that you saw a squirrel) and disregard individual information (e.g. whether that squirrel had a long or short tail). Across two experiments with preverbal infants, we demonstrated that it is not language but nonverbal category knowledge that determines what information is packed into object representations. Twelve-month-olds ($N = 48$) participated in an electroencephalography (EEG) change-detection task involving objects undergoing a brief occlusion. When viewing objects from unfamiliar categories, infants detected both across- and within-category changes, as evidenced by their negative central wave (Nc) event-related potential. Conversely, when viewing objects from familiar categories, they did not respond to within-category changes, which indicates that nonverbal category knowledge interfered with the representation of individual surface features necessary to detect such changes. Furthermore, distinct patterns of γ and α oscillations between familiar and unfamiliar categories were evident before and during occlusion, suggesting that categorization had an influence on the format of recruited object representations. Thus, we show that nonverbal category knowledge has rapid and enduring effects on object representation and discuss their functional significance for generic knowledge acquisition in the absence of language.

# 1. Introduction

We filter experience through knowledge about the world such that we cannot help recognizing things around us as tomatoes or apples, pens or pencils, cars or computers [1]. This structuring of perceptual input through abstract semantic categories bears heavily on the way we represent and remember encountered objects. Categorical identity takes priority over featural information in visual processing: it is easier to discriminate objects drawn from distinct categories than from the same category even when the visual distance between the items to compare is matched (e.g. B–p are easier to tell apart than B–b, [2–4]). Categorical information (e.g. whether one saw a lamp or a chair) is also easier to recall than individual information (e.g. which particular lamp was seen), whether from short- or long-term memory [5–7].

Developmental studies indicate that categorization starts to influence object representation already during the first year of life. Like adults, infants privilege category over featural information when individuating objects and storing them in working memory [8–13] even though they are capable of encoding visual features such as shape, size, pattern and colour [14]. To date, however, the origins of biases prioritizing categorical information at the expense of individual features remain unclear despite their apparent continuity throughout the lifespan.

According to one influential tradition, categorical information holds a special role in object representation because of language that guides category learning [15,16]. As early as three months of age, infants form abstract visual categories by extracting commonalities among disparate individuals named in the same way [17], and language continues to facilitate the discovery of new categories into adulthood [18]. In addition, both adults [19] and children [20] tend to retrieve category labels (e.g. 'apple', 'cat') upon seeing familiar objects. As a result, some suggested that top-down influences from lexical-semantic representations to higher-level visual areas bring out the categorical features of perceived objects, inducing biases in perception and memory [21]. Biases may also result from the fact that words are held in memory instead of visual representations [22–25]. Indeed, in adults, when lexical access is prevented through verbal interference, the advantages for encoding category information become attenuated [26,27].

However, language itself builds on pre-existing conceptual representations that become word meanings in the course of language acquisition [28–30]. Infants map novel labels not onto the particular objects they experience being named but directly onto prelinguistic concepts and categories that these objects exemplify [31,32]. Critically, for such mappings to take place, learners need to be equipped with prelinguistic representational capacities that make conceptual category information stand out before linguistic representations become available [33]. This opens the possibility that nonverbal knowledge structures that predate language may be responsible for the prioritization of categorical information in object representation.

Does the priority of categorical information in object representation reflect biases stemming from language or from non-linguistic concepts and categories? The available experimental evidence cannot adjudicate between these two possibilities because it comes predominantly from studies that used lexicalized categories, thus leaving unanswered whether categorical biases generalize to non-lexicalized categories. Besides, comparing the effects of lexicalized and non-lexicalized categories in adults is problematic because of our tendency to spontaneously label unfamiliar objects involved in the task at hand [3]. Data from preverbal infants could provide insight into the origins of categorical biases, but the available experimental work investigating infant object cognition has not systematically controlled for the contributions of infant lexical knowledge, shown to be operational as early as six to nine months of age [34,35].

Using a novel change-detection task in a population of 12-month-olds, the present study investigated whether nonverbal categories can modify how object representations are set up in the absence of language. We employed electroencephalography (EEG) to assess two related issues. First, we enquired whether newly taught nonverbal category knowledge could bias infants to focus on categorical features of viewed items and give up representing their individual features. To probe what information infants encode, we compared their neural responses to two types of object changes occurring after a short occlusion of a probe item: within-category changes (e.g. a watering can changing into another watering can) and across-category changes (e.g. a watering can changing into a padlock). While across-category changes can be detected based on categorical information only, the detection of within-category changes requires the storage of individual features. Thus, if nonverbal category knowledge enhances the representation of categorical attributes at the expense of individual detail, this should be reflected in the infants' sensitivity to object changes: namely, infants should manifest higher sensitivity to across- than within-category changes.

To assess change detection performance, we measured the negative central wave (Nc) in response to object reappearance. The Nc is an event-related component observed in the infant EEG at frontocentral sites following modifications in stimulus appearance, reflecting the allocation of attention ([36–39]; for a review, [40]). Hence, differences in Nc amplitudes between conditions would indicate differences in the infants' attentional engagement. In our design, the only aspect of the stimulus that varied across conditions and could modulate infants' attention was the identity of the object revealed after occlusion. Therefore, we reasoned that the Nc component would index the sensitivity to different object changes (i.e. with larger amplitudes diagnostic of larger sensitivity) and provide a window into the contents of the underlying object representations. To verify that the observed patterns of results were brought about by category knowledge and would not arise spontaneously in the absence thereof, we contrasted infants' responses to objects from familiar and unfamiliar categories.

Second, we sought to evaluate whether nonverbal categorization affected the format of representations recruited by the infants and shed light on the time course with which this process may occur. For example, one possibility is that, upon categorization, infants have accessed and stored a nonverbal symbol corresponding to the category of a viewed object [33,41]. Alternatively, categorization may only have affected the selection of visual features infants encoded and retained in memory (e.g. category-diagnostic features versus a random selection of features including category-diagnostic features and other features irrelevant for categorization). These changes could occur already upon visual inspection of the objects or, only later, when the objects became occluded. To inquire whether categorization might be triggering a representational change, we investigated induced oscillatory activity before object reappearance. Previous infancy work revealed that induced γ oscillations over temporal cortices exhibit higher synchrony during a visual inspection of objects from familiar compared with unfamiliar categories [42], which was interpreted as an index of conceptual access. We sought to replicate this pattern of findings. However, scalp-recorded γ activation may reflect differences in eye movements rather than in neural representations [43,44]. We, therefore, chose to additionally investigate another neural marker, α-band oscillations, widely linked to the maintenance of sensory information in working memory (for a review, [45]). We posited that if posterior α suppression during occlusion is manifest only for unfamiliar categories, this would indicate that distinct representational formats might have been recruited, with unfamiliar-category objects represented by their features actively maintained in the ventral visual stream and familiar-category objects potentially represented in a categorical or conceptual symbolic manner with lesser involvement of ventral visual cortices [46,47].

We carried out two experiments. To validate our design, in Experiment 1, we sought to confirm that our EEG paradigm is sensitive to the existence of categorical biases in object representation, thus replicating prior behavioural findings. We tested whether infants would prioritize categorical over individual information when tracking through occlusion objects from real-world categories: familiar, acquired before the laboratory visit and unfamiliar. This design capitalized on the conceptual repertoire that infants develop during the first year of life, which partly overlaps with their receptive lexicon [34,35], therefore, leaving open the possibility that the observed effects might be due to early linguistic experience. To rule this out and directly assess the role of nonverbal category knowledge, in Experiment 2, we investigated strictly nonverbal categories taught during the experimental session. By introducing novel categories in the laboratory, we ensured that infants did not know their names and, thus, could not recruit their lexical knowledge to support object representation.

# 2. Experiment 1

## 2.1. Material and method

### 2.1.1. Participants

Data were obtained from 24 healthy 12-month-olds from monolingual English-speaking homes: 12 infants participated in the familiar-category condition (five females, mean age = 12.76 months, range: 12.38–13.03 months) and 12 in the unfamiliar-category condition (seven females, mean age = 12.66 months, range: 12.23–13.06 months). The current sample size was determined *a priori* based on the literature [39,42,48,49]. Within-subject comparisons of mean Nc amplitudes yield very large effect sizes ($d > 1.20$, [48]). Hence, using power analysis in G*Power 3.1 [50] we estimated that testing 12 participants per condition would be sufficient to provide 80% statistical power to detect a large effect size ($d = 0.80$) on the Nc responses using a paired-samples $t$-test (one-sided) and an α of 0.05. Note also that in across-

experiment comparisons the sample size for each category (familiar versus unfamiliar) raises to 24, therefore, providing 95% statistical power to detect a large effect size using a two-sided test. Another 18 infants were tested but not included in the analysis because of fussiness ($n = 4$) or excessive movement resulting in an insufficient number of artefact-free trials ($n = 14$). This attrition rate conforms to prior work in infant electrophysiology [51–53]. Families were recruited through advertising in local magazines. Before the experimental session, all carers gave written informed consent. Families' travel expenses were reimbursed. Infants were rewarded for their participation with a certificate and a small gift.

### 2.1.2. Stimuli

In the familiar-category condition, we used six object kinds whose names were previously shown to be comprehended by 12-month-olds growing up in monolingual English-speaking families (for questionnaire evidence, [54,55]; for experimental evidence: e.g. [34]): BALL, BOTTLE, CAR, DUCK, SHOE, TEDDY BEAR. Carers confirmed the familiarity with the category labels in the current sample through the MacArthur Communicative Development Inventory: Words and Gestures [56,57]. Average comprehension scores ranged from 65% to 95% (ball: 95%, bottle: 83%, car: 91%, duck: 78%, shoe: 84%, teddy bear: 65%). In the unfamiliar-category condition, we used six objects whose kinds and labels were unknown to the infants: FEATHER, GUITAR, HEDGEHOG, PADLOCK, STAPLER, WATERING CAN. Parents were shown the stimuli before the EEG session and confirmed that infants were not familiar with any of the target categories.

Visual stimuli in the EEG task were temporarily occluded colour photographs of real-world objects. For each object kind, we used photographs of three different items. Whenever possible, we chose items of different colour, orientation and shape to maximize within-category variability.

Our stimuli are freely available in the following OSF repository: https://osf.io/b36cg/.

### 2.1.3. Design and procedure

Infants watched animations depicting objects presented against a white background and briefly occluded by a vertically moving screen. Each trial began with an empty stage (300–550 ms). Then, an object (e.g. duck A) slid onto a table (300 ms) and remained in full view (1000 ms) before disappearing behind a rising screen (350 ms). The screen hid the object for 1000–1250 ms (randomly determined) and then went down (350 ms), revealing one of the three possible outcomes: either the same object that was occluded (e.g. duck A on no change trials), or another exemplar of the same category (e.g. duck B on within-category change trials), or an exemplar of a different category (e.g. shoe on across-category change trials). The object briefly stayed in full view (1000 ms) before sliding down behind the stage (300 ms). The three conditions (no change, within-category change, across-category change) were presented with the same frequency (i.e. on one-third of all trials each). The order of the presentation was pseudorandomized, as no more than two trials from the same condition could appear in succession. The trials were presented in silence, only the object sliding up on the stage was indicated by a short jingle (200 ms, randomly selected from a set of three sounds) whose onset was timed to the first frame in which the object became visible. Infants were randomly assigned to one of the two conditions: familiar-category condition involving tokens of object categories familiar to them (BALL, BOTTLE, CAR, DUCK, SHOE, TEDDY BEAR) and unfamiliar-category condition involving tokens of unfamiliar object categories (FEATHER, GUITAR, HEDGEHOG, PADLOCK, STAPLER, WATERING CAN).

Infants were seated on their carers' lap in a dimly lit, electrically shielded and sound-proofed room. Carers were instructed not to name the stimuli. After fitting the EEG cap, the experimenter pointed to the stimulus screen and talked briefly to the infant, while an attention-getter animation was displayed (i.e. flowers opening up to the sound of ambient music), then left the room. The data collection was carried out for as long as the infants were willing to participate. The experimenter occasionally called the infant's name through a loudspeaker and praised her for attending to the stimuli (e.g. 'Hi INFANT'S NAME! You're doing a great job. Look at these. Wow! What are they?'). If the experimenter judged necessary to redirect infants' attention back to the stimuli presentation, an attention-getter clip (i.e. a young woman saying 'Hello baby!' while waving to the infant) was played. The study protocol was prepared in compliance with relevant ethical regulations and approved by the ethical commission of Birkbeck College, University of London.

MATLAB (7.10.0.499) and Psychophysics Toolbox (3.0.8) were used for stimuli randomization and presentation. Stimuli ($800 \times 600$ pixels) were displayed on a CRT screen (100 Hz refresh rate). The viewing distance was approximately 80 cm from the display monitor. From this distance, the $15 \times 15$ cm stimuli subtended 11° of visual angle.

## 2.1.4. Data analysis

### 2.1.4.1. EEG data acquisition and preprocessing

Continuous EEG was acquired using HydroCel Geodesic Sensor Nets composed of 124 sensors (EGI, Eugene, OR, USA). Recordings were referenced to the vertex (Cz in the 10–20 system). The ground electrode was at the rear of the head (between Cz and Pz). The data were bandpass filtered online at 0.1–200 Hz and sampled at 500 Hz. Because infants sometimes look away from the display, the EEG data were first coded for visual attendance based on time-locked video recordings. All events (i.e. object presentation and occlusion; object reappearance after occlusion) during which infants did not attend to the screen were manually excluded from the analysis based on video inspection. Additionally, object reappearance events were excluded if infants did not look at the screen before occlusion because that rendered impossible encoding the object and change detection. The continuous EEG was offline bandpass filtered at 0.3–100 Hz (following [35]). Although filtering can distort the time course and amplitude of the event-related potential (ERP) waveforms, such distortions typically occur when the low cut-off exceeds 0.5 Hz or when the high cut-off is below 10 Hz [58,59].

### 2.1.4.2. Derivation of event-related potentials

The continuous EEG signal was segmented into 1700 ms epochs, beginning 200 ms before the first frame when the occluder started to go down (considered Time 0). Trials were sorted according to the occlusion outcome (i.e. no change, within-category change or across-category change). Automatic artefact detection and visual inspection were applied to identify channels contaminated by noise: channels were automatically marked as bad if they contained eye movements (i.e. whenever the average amplitude of an 80 ms gliding window exceeded 55 µV at horizontal electrooculography (EOG) channels or 140 µV at vertical EOG channels) or if they were contaminated by body movements (i.e. the average amplitude of an 80 ms gliding window exceeded 200 µV at any channel). Segments were automatically removed if they contained ocular artefacts or if more than 10% of the channels were contaminated by movement artefacts. For the remaining segments, bad channels were interpolated using spherical spline interpolation. For each participant, the data were then baseline-corrected to the first 200 ms of the segment, averaged for each trial type and re-referenced to the average reference. Each infant was required to contribute at least 10 trials to each condition to be included in the final analysis. As the Nc component is well described in time and space, we determined the specific cluster of electrodes and time window of analysis based on the prior findings showing a broad central to the anterior distribution of the Nc (spanning between Cz and Fz, e.g. [38,39]) and computed the mean Nc amplitude using a cluster of electrodes covering central and anterior sites (3, 4, 5, 6, 7, 10, 11, 12, 13, 16, 18, 19, 20, 23, 24, 28, 29, 30, 35, 36, 104, 105, 106, 110, 111, 112, 117, 118, 124) between 350 and 800 ms following the reappearance of the object (i.e. relative to Time 0 of the segment). *T*-tests below are two-tailed. Additionally, we report 95% confidence intervals (CI) for the difference in the mean of the dependent variable and Cohen's *d* for effect size.

On average, infants contributed 13.2 (s.d. = 2.1) artefact-free segments (familiar-category condition: $M = 13.3$, s.d. = 2.5, $R = 10$–19, no change: $M = 13.7$, s.d. = 2.5, within-category change: $M = 13.71$, s.d. = 2.8, across-category change: $M = 13.1$, s.d. = 2.4; unfamiliar-category condition: $M = 13$, s.d. = 1.6, $R = 10$–17, no change: $M = 13.1$, s.d. = 1.8, within-category change: $M = 13.2$, s.d. = 1.5, across-category change: $M = 12.8$, s.d. = 1.7). There were no significant differences in the number of artefact-free segments between occlusion outcomes within- or across conditions, all $p$s > 0.50. Therefore, signal-to-noise levels across conditions were similar. Similarly, no significant differences were recorded in the overall numbers of segments before artefact detection (all $p$s > 0.28; familiar-category condition: $M = 23.9$, s.d. = 3.8, $R = 17$–32, no change: $M = 24.2$, s.d. = 4.2, within-category change: $M = 24.5$, s.d. = 4.1, across-category change: $M = 23.1$, s.d. = 3.4; unfamiliar-category condition: $M = 25.7$, s.d. = 6.46, $R = 13$–39, no change: $M = 26.5$, s.d. = 7.3, within-category change: $M = 25.5$, s.d. = 5.8, across-category change: $M = 25.2$, s.d. = 6.7 s). This indicates that infants presented with familiar and unfamiliar categories paid equal attention to the stimuli.

## 2.2. Results and discussion

To assess infants' sensitivity to object changes, we compared the Nc response to object reappearance on trials in which the identity of the objects was maintained throughout occlusion and trials on which it changed, across familiar and unfamiliar categories (figure 2). The average Nc amplitudes were

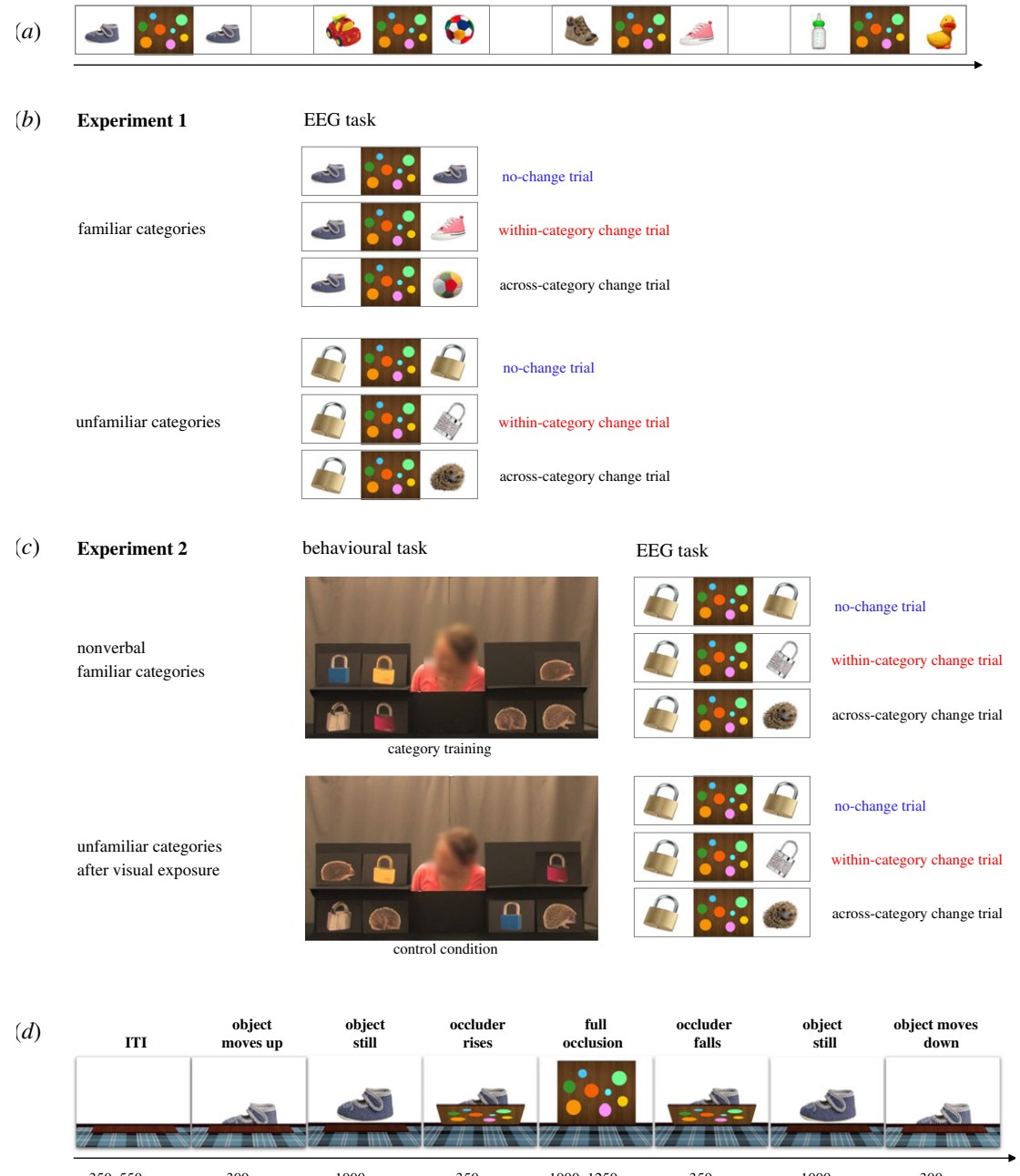

**Figure 1.** Schematic of experimental design. (*a*) EEG change-detection task. In the EEG change-detection task employed across Experiments 1 and 2, infants watched a series of briefly occluded objects. One of the three outcomes was presented after occlusion: (i) no change to the object that was occluded, (ii) within-category change with a novel item of the same category replacing the initial object, (iii) across-category change with a novel item from a different category replacing the initial object. We varied whether infants watched familiar or unfamiliar categories. (*b*) Schematic of Experiment 1. Two groups of infants participated in the EEG task: infants in one group watched objects from familiar categories learnt prior to the laboratory visit and infants in the other group watched objects from unfamiliar categories. (*c*) Schematic of Experiment 2. The EEG task was preceded by a behavioural task: one group of infants participated in a nonverbal category training, while the other group was shown the same visual stimuli in a manner not conducive to category formation (control condition). The final frame of videos used in respective behavioural tasks is presented in the display. The same picture stimuli were used as in the unfamiliar-category condition of Experiment 1, hence the only difference between groups was the category knowledge induced in the behavioural task. (*d*) Time course of an EEG trial. The EEG trial structure was identical across all experiments and conditions.

entered into a mixed-model ANOVA with occlusion outcome (no change versus within-category change versus across-category change) as a within-subject factor and condition (familiar category versus unfamiliar category) as a between-subject factor. This analysis yielded a significant main effect of

outcome, $F_{2,44} = 21.583$, $p < 0.001$, $\eta_p^2 = 0.50$, and a significant interaction between outcome and condition, $F_{2,44} = 7.020$, $p = 0.002$, $\eta_p^2 = 0.24$. Infants in both conditions proved sensitive to object changes that occurred after occlusion (familiar-category condition: $F_{2,22} = 20.122$, $p < 0.001$, $\eta_p^2 = 0.65$; unfamiliar-category condition: $F_{2,22} = 9.233$, $p = 0.001$, $\eta_p^2 = 0.46$, as shown by separate repeated-measures ANOVAs within each category condition), but they reacted differently to within-category changes. Follow-up $t$-tests revealed that infants who viewed familiar categories noted across-category object changes, as evidenced by a more negative Nc response on across-category change trials than no change trials, $t_{11} = 5.552$, $p < 0.001$, 95% CI = [3.59, 8.31], $d = 1.60$, but they failed to display sensitivity to within-category object changes, as shown by the comparable responses on within-category change and no change trials, $t_{11} = 0.597$, $p = 0.562$, 95% CI = [−1.27, 2.21], $d = 0.17$ (within-category versus across-category change trials, $t_{11} = 4.490$, $p < 0.001$, 95% CI = [2.79, 8.16], $d = 1.30$). By contrast, infants presented with unfamiliar categories detected changes in objects' identity regardless of whether the post-change object came from the same or a different category than the initial one. Compared with no change trials, their Nc was significantly more negative on both within-category change trials, $t_{11} = 3.577$, $p = 0.004$, 95% CI = [1.63, 6.83], $d = 1.03$, and across-category change trials, $t_{11} = 3.260$, $p = 0.008$, 95% CI = [1.33, 6.84], $d = 0.94$. There was no significant difference between within- and across-category object changes, $t_{11} = 0.163$, $p = 0.874$, 95% CI = [−2.08, 1.80], $d = 0.05$.

These results indicate that the response to within-category changes was affected by category knowledge. Infants who viewed familiar categories responded selectively to across-category but not within-category object changes, while those who viewed unfamiliar categories responded reliably to both kinds of changes. The attenuated sensitivity to within-category changes occurring in objects drawn from familiar categories suggests that category knowledge impeded the representation of individual detail irrelevant for categorization. These results conceptually replicated previous behavioural findings [8–11] and confirmed that categorical biases in object representation are manifest at the electrophysiological level. We could thus proceed to test whether nonverbal category knowledge alone would similarly modify infant object representation.

Note, however, that although we sought to select items that, upon visual inspection, would be equally perceptually distinct from each other, the stimuli in the unfamiliar-category condition might have been more heterogeneous than in the familiar-category condition. Thus, the current pattern of results might have been due to the higher category discriminability within the unfamiliar compared with familiar categories. This issue was also addressed in Experiment 2.

# 3. Experiment 2

We first taught infants two novel categories without using category labels, and then assessed whether this newly acquired nonverbal category knowledge had an impact on their object representations. That is, the EEG task was preceded by a short laboratory-based category training (category-training condition) previously shown to induce category learning in young infants [60]. To directly compare Experiments 1 and 2, and to confirm that the effects in Experiment 1 were due to category knowledge rather than physical characteristics of the stimuli (i.e. larger perceptual distance between category tokens in the unfamiliar categories boosting within-category change detection), we used the same visual stimuli as in the unfamiliar-category condition of Experiment 1.

## 3.1. Material and method

### 3.1.1. Participants

Twenty-four healthy monolingual infants from English-speaking homes took part in this experiment: 12 were assigned to the category-training condition (six females, mean age = 12.64 months, range: 12.29–13.03 months) and 12 to the familiarization condition (five females, mean age = 12.65 months, range: 12.16–12.97 months). A further 23 infants were tested but not included in the analysis because of fussiness ($n = 5$) or excessive movement resulting in an insufficient number of artefact-free trials ($n = 18$). As in Experiment 1, families were recruited through advertising in local magazines, and their travel expenses were reimbursed. All carers gave written informed consent. Infants were rewarded with a certificate and a small gift.

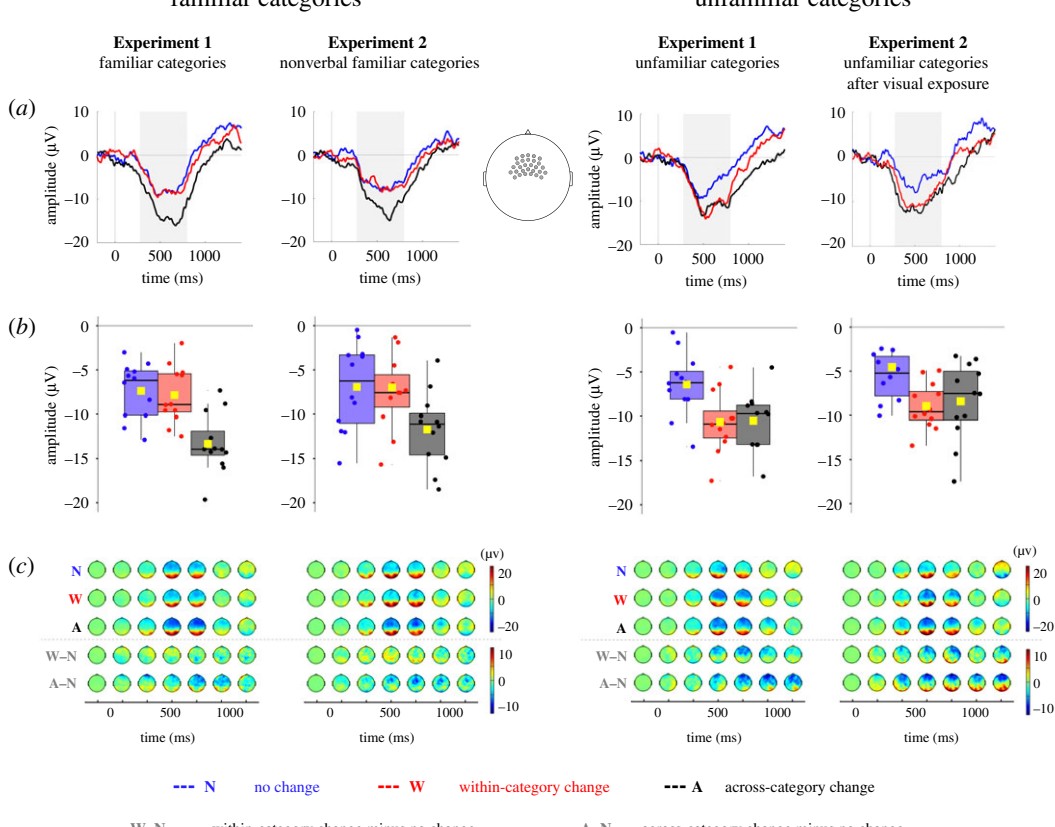

**Figure 2.** Infants' change-detection performance in Experiments 1 and 2. (*a*) Grand-average waveforms represent the Nc component in response to the object's reappearance after occlusion on no change trials (blue), within-category change trials (red) and across-category change trials (black). The grey shading indicates the time window of the analysis (350–800 ms). The vertical grey line marks the time at which the object became first visible after occlusion (time 0). (*b*) Average Nc amplitude across object-change conditions. Yellow squares indicate means. Black horizontal lines indicate medians. The bottom and the top of the boxes represent the first and the third quartiles. Whiskers extend from the middle quartiles to the smallest and largest values within 1.5 times the interquartile range. Dots represent individual data points. (*c*) Event-related potential topographies over trial time for each object-change condition (N, no change; W, within-category change; A, across-category change) and their difference (W−N, within-category change minus no change; A−N, across-category change minus no change). Voltages were averaged within 200 ms bins.

### 3.1.2. Stimuli, design and procedure

The EEG task of an identical structure as in Experiment 1 was directly preceded by a short behavioural procedure: category training or control (using the same visual stimuli and matched for exposure time, but not conducive to category learning), administered in a different experimental room than the EEG task. Because it is unknown whether infants at this age can readily learn more than two categories in a laboratory setting, we limited the number of taught categories to two. Individual infants were behaviourally trained and then tested in the EEG task on one of the following pairs: FEATHER and WATERING CAN, GUITAR and STAPLER, HEDGEHOG and PADLOCK. Each pair was presented to four participants from the final sample. Four additional photographs for each category, different from the ones used for the EEG recording, were used during the behavioural task.

During category training, infants watched videos of an experimenter sorting category exemplars (depicted on picture cards) into two locations based on their category membership (e.g. watering cans placed to the left versus staplers placed to the right, forming two category-based homogeneous object sets). In each trial, an actress was presented behind a table with two shelves, one on the right, containing three exemplars of category A, and one on the left, containing three exemplars of category B. First, the actress greeted the infant (*'Hello baby!'*) while waving and smiling at her. Then, she retrieved from behind the table a picture card representing a new category token, presented it to the infant saying *'Look at this!'*, placed it on one of the two shelves and remained immobile looking in the direction of the placed object (see figure 1). The actress always sorted the new

picture in agreement with the category of the represented object, while the side of placement (left versus right) was counterbalanced. Images of the target items were glued onto black pieces of cardboard and used to make video stimuli. Trials were separated by a short centrally displayed attention getter. There were four trials per category, for a total of eight videos. One trial lasted 15 s, for a total duration of the session of approximately 2 min. This training procedure, modelled on previous work in infant category learning [60], was validated in a behavioural pilot using a looking-time violation-of-expectation test and administered to a separate group of participants (see the electronic supplemental material: ESM1. Category-training procedure. Task design and behavioural pilot data).

Half of the infants participated in the category training. The other half received the same amount of exposure to the exemplars of novel categories, but in a manner not conducive to category formation (control condition, [31,61]): the experimenter sorted objects in a non-systematic way, producing two mixed-category object sets (each containing some instances of staplers and watering cans). The trial structure and timing were the same as in the category training, but each of the two shelves contained a mixed set of items from both categories. This condition was designed to control for the effects of prolonged perceptual exposure to unfamiliar objects.

### 3.1.3. Data analysis

The EEG data were acquired and analysed following the same processing and analysis steps as in Experiment 1. The same electrode cluster and time window were used for the Nc derivation. Overall, infants contributed 13.7 (s.d. = 2.7) artefact-free segments (category training: $M = 13.8$, s.d. = 2.9, $R =$ 10–21, no change: $M = 14.2$, s.d. = 3.5, within-category change: $M = 13.8$, s.d. = 2.4, across-category change, $M = 13.4$, s.d. = 2.9; control: $M = 13.5$, s.d. = 2.4, $R = 10$–20, no change: $M = 14.1$, s.d. = 2.8, within-category change: $M = 13.2$, s.d. = 1.9, across-category change, $M = 13.2$, s.d. = 2.6). The number of artefact-free segments did not differ across occlusion outcomes within and across groups, all $p$s > 0.17, indicating a comparable signal-to-noise levels across conditions. The overall number of segments infants saw was comparable across occlusion outcomes ($p$s > 0.32), but varied across groups with infants in the category-training condition having seen on average more segments than infants in the control condition ($p = 0.004$; category training: $M = 31.4$, s.d. = 8.3, $R =$ 11–45; no change: $M = 31.8$, s.d. = 8.2; within-category change: $M = 31.5$, s.d. = 8.6; across-category change: $M = 31.1$, s.d. = 8.9; control: $M = 22.4$, s.d. = 5, $R = 13$–34, no change: $M = 22.7$, s.d. = 5.8, within-category change: $M = 22.6$, s.d. = 4.8, across-category change: $M = 23.1$, s.d. = 4.4). This suggests that infants in this experiment sustained their attention to familiar categories for longer than to unfamiliar categories.

All participants who provided enough artefact-free trials to be included in the final EEG sample watched the entirety of the category training or the matched control procedure administered before the EEG task.

## 3.2. Results and discussion

As in Experiment 1, the average amplitudes of the Nc component elicited by object reappearance after occlusion (depicted in figure 2) were entered into a mixed-model ANOVA, with occlusion outcome (no change versus within-category change versus across-category change) as a within-subject factor and condition (category training versus control condition) as a between-subject factor. This analysis revealed a significant main effect of outcome, $F_{2,44} = 20.688$, $p < 0.001$, $\eta_p^2 = 0.46$, and a significant interaction between occlusion outcome and condition, $F_{2,44} = 8.770$, $p = 0.001$, $\eta_p^2 = 0.29$. This interaction was further explored by separate one-way ANOVAs within each condition, confirming that both infants who participated in the category training and those who participated in the control condition were sensitive to changes in occlusion outcomes (category-training condition: $F_{2,22} = 15.375$, $p < 0.001$, $\eta_p^2 = 0.58$; control condition: $F_{2,22} = 13.973$, $p < 0.001$, $\eta_p^2 = 0.56$).

The results of follow-up tests mirrored the results obtained in Experiment 1. Infants who learned nonverbal categories prior to the EEG task displayed sensitivity to across-category but not to within-category object changes, exhibiting the same pattern of Nc response as infants who acquired category knowledge prior to the laboratory visit. This was demonstrated by a significantly more negative Nc on across-category change trials relative to no change trials, $t_{11} = 4.030$, $p = 0.002$, 95% CI = [2.16, 7.36], $d = 1.16$, or with category change trials, $t_{11} = 5.409$, $p < 0.001$, 95% CI = [2.80, 6.64], $d = 1.56$, and a comparable response on within-category change and no change trials, $t_{11} = 0.050$, $p = 0.961$, 95% CI = [−1.88, 1.97], $d = 0.01$. On the other hand, infants in the control condition, similarly to infants

presented with unfamiliar categories in Experiment 1, detected both across-category and within-category object changes, as indicated by a significantly more negative Nc wave on across-change trials, $t_{11} = 4.292$, $p = 0.001$, 95% CI = [1.89, 5.88], $d = 1.68$, and within-change trials, $t_{11} = 5.835$, $p < 0.001$, 95% CI = [2.75, 6.09] $d = 1.24$, than on no change trials. Responses in both change conditions did not differ from each other, $t_{11} = 0.511$, $p = 0.620$, 95% CI = [−2.85,1.78], $d = 0.15$.

This pattern of responses indicates that in-laboratory induction of nonverbal category knowledge, but not mere visual experience with exemplars of unfamiliar categories, compromised within-category discrimination. The fact that infants in category training and familiarization conditions viewed the same visual stimuli and differed only in their access to category knowledge demonstrates that their change-detection performance was not an artefact of our visual stimuli selection. Rather, upon learning nonverbal visual categories, infants selectively focused on the information related to category membership, seemingly leaving other information such as non-categorical visual features out. Importantly, this process took place in the absence of language. Thus, our results provide evidence that lexical knowledge is not necessary for the categorical biases to modulate visual object representations. The availability and recruitment of nonverbal category structures alone limits the number of visual features that are contained in object representation.

## 3.3. Comparison of Experiments 1 and 2

A mixed-model ANOVA with occlusion outcome (no change versus within-category change versus across-category change) as a within-subject factor, and category knowledge (familiar versus unfamiliar category) and experiment (Experiment 1 versus Experiment 2) as between-subject factors, revealed a significant main effect of outcome, $F_{2,88} = 42.120$, $p < 0.001$, $\eta_p^2 = 0.49$, and a significant interaction between outcome and category knowledge, $F_{2,88} = 15.448$, $p < 0.001$, $\eta_p^2 = 0.26$. The lack of a significant main effect of experiment, $F_{1,44} = 2.478$, $p = 0.123$, $\eta_p^2 = 0.05$, or significant interactions with this factor (all $p$s > 0.620) suggests that the presence of category knowledge, whether long-term (Experiment 1) or newly acquired (Experiment 2), had the same effect on the infants' sensitivity to object changes. Additionally, there was no evidence that simple visual exposure to unknown objects before the EEG test (Experiment 2) had an effect on the neural activity elicited by the unfamiliar categories. The data collapsed across experiments yielded the same patterns of results as the analyses within each experiment: within-category change detection was compromised for familiar categories, $t_{23} = 0.446$, $p = 0.660$, 95% CI = [−0.94, 1.45], $d = 0.09$, but not for unfamiliar categories, $t_{23} = 6.296$, $p < 0.001$, 95% CI = [2.90, 5.74], $d = 1.28$, as revealed by the comparisons between within-category change and no change conditions. Conversely, sensitivity to across-category changes was manifest for both familiar categories, $t_{23} = 6.781$, $p < 0.001$, $d = 1.38$, 95% CI = [3.72, 6.99], and unfamiliar categories, $t_{23} = 5.271$, $p < 0.001$, $d = 1.08$, 95% CI = [2.42, 5.55], by comparisons with no change.

# 4. Time course and nature of categorical biases on object representation

While the Nc results provide evidence for modulatory effects of nonverbal category knowledge on object representation, they cannot indicate at what stage of object processing the categorical biases emerge and whether they affect only the selection of features stored in the representation or, additionally, the format of this representation. To address these questions, we analysed event-related induced oscillatory activity in the γ and α ranges. γ-band oscillations have been previously associated with object processing in human adults [62], human infants [42,51,52,63,64] and non-human animals [65]. α-band oscillations have been argued to regulate working-memory representations [45,66,67].

## 4.1. Time–frequency analysis

The continuous EEG signal was segmented into 4700 ms epochs, beginning 1050 ms before the object started to be visible on the stage. Trials were sorted according to familiarity with the presented category (familiar versus unfamiliar). We applied the same artefact and bad channel interpolation routine as in the ERP analysis. To compute induced event-related oscillations, we followed an established routine [63]: the continuous wavelet transformation was applied to individual artefact-free epochs, using Morlet wavelets with 1 Hz resolution in the range of 5–60 Hz (EEGLAB, v. 9.0.5.6b; and EEGLAB-based scripts to perform the wavelet transform, available online, see [35]). We took the absolute output value (i.e. the amplitude). Next, baseline correction was performed by subtracting the

average activity during 200 ms immediately preceding the object appearance (i.e. corresponding to the phase of the trial when the stage was empty) from the whole epoch at each frequency; 500 ms at the beginning and the end of each epoch were removed to eliminate the distortion created by the wavelet transform. Average wavelet coefficients within infants were calculated by taking the mean across trials for each time window and frequency band of interest.

Time–frequency activation was averaged within two frequency ranges (γ range: 25–45 Hz, α range: 6–9 Hz) at two electrode sites over posterior temporal cortex (left: 58, 59, 64, 65, 66, 69, 70; right: 83, 84, 89, 90, 91, 95, 96), selected based on the literature [42,51,64,68]. The first time window corresponded to the full visibility of the target object (*presentation phase:* 300–1300 ms) and the second to its full occlusion (*occlusion phase:* 1650–2650 ms). All *t*-tests reported below are two-tailed. We also report 95% CI for the difference in the mean of the dependent variable and Cohen's *d* for effect size.

There was no difference in the number of artefact-free trials contributed by the infants across conditions, with neither category knowledge (familiar versus unfamiliar), $F_{1,44} = 0.613$, $p = 0.438$, nor experiment (1 versus 2), $F_{1,44} = 0.316$, $p = 0.577$, affecting the number of valid segments (familiar category in Experiment 1: $M = 27$, $R = 10$–57; unfamiliar category in Experiment 1: $M = 22$, $R = 11$–37; category training in Experiment 2: $M = 23$, $R = 11$–53; control in Experiment 2: $M = 23$, $R = 10$–51). There was a significant interaction between experiment and category knowledge, $F_{1,44} = 16.309$, $p < 0.001$, when we considered the overall numbers of segments infants saw, as assessed before the exclusion of trials contaminated by artefacts. There were no significant main effects of experiment, $F_{1,44} = 0.123$, $p = 0.727$, or category, $F_{1,44} = 2.428$, $p = 0.126$. In Experiment 1, infants in the familiar-category condition saw a comparable amount of segments ($M = 87$, s.d. = 8) to those who watched unfamiliar categories ($M = 99$, s.d. = 20), $p = 0.057$. In Experiment 2, infants in the familiar-category condition watched overall more segments ($M = 109$, s.d. = 25) than those in the unfamiliar-category condition ($M = 80$, s.d. = 14), $p = 0.003$.

## 4.2. Results and discussion

The results are depicted in figure 3. The activations in the γ and α ranges were entered into mixed-model ANOVAs with category knowledge (familiar versus unfamiliar) and experiment (Experiment 1 versus Experiment 2) as between-subject factors, phase (presentation versus occlusion) and hemisphere (left versus right) as within-subject factors.

The ANOVA in the γ range (25–45 Hz) revealed two effects that approached significance: an interaction between experiment, category and phase, $F_{1,44} = 2.818$, $p = 0.100$, $\eta_p^2 = 0.06$, and a main effect of category knowledge, $F_{1,44} = 2.772$, $p = 0.106$, $\eta_p^2 = 0.06$. Because previous findings suggested that in 12-month-olds γ-band activity might be an index of conceptual processing during visual inspection of familiar objects [69], we carried out exploratory analyses to probe whether such modulation was manifest also in the present data. Separate ANOVAs with category knowledge and experiment were conducted within each phase (presentation versus occlusion). The data were collapsed across the bilateral clusters of electrodes, as there was no significant main effect of hemisphere nor any interaction with this factor (all *p*s > 0.231). During the presentation, γ synchrony was only modulated by category knowledge, $F_{1,44} = 4.520$, $p = 0.039$, $\eta_p^2 = 0.09$, with familiar categories eliciting higher activation than the unfamiliar ones. Comparisons with baseline revealed that only the response to familiar categories raised significantly above baseline, $t_{23} = 3.754$, $p = 0.001$, $d = 0.77$, 95% CI = [0.03, 0.12], unfamiliar categories: $t_{23} = 0.562$, $p = 0.579$, $d = 0.11$, 95% CI = [−0.03, 0.06]. This pattern of activation replicates previous findings that linked this frequency band to conceptual processing of familiar categories in young infants [42] and is consistent with the idea that category knowledge had an impact on object representation already at the stage of information encoding. Future research should further validate this possibility. The ANOVA on the occlusion data did not reveal any significant effects (all *p*s > 0.350).

The omnibus ANOVA in the α range (6–9 Hz) yielded a significant main effect of phase, $F_{1,44} = 19.557$, $p < 0.001$, $\eta_p^2 = 0.31$, and a significant interaction between category knowledge, experiment, hemisphere and phase, $F_{1,44} = 4.584$, $p = 0.038$, $\eta_p^2 = 0.09$. To resolve this interaction, we conducted separate ANOVAs within each phase, with category knowledge (familiar versus unfamiliar) and experiment (Experiment 1 versus Experiment 2) as between-subject factors, and hemisphere (left versus right) as a within-subject factor. The ANOVA on the presentation data yielded no significant effects (all *p*s > 0.189). A comparison with baseline performed on the data collapsed across categories, hemispheres and experiments revealed that the α activation was suppressed, that is significantly lower than at baseline, $t_{47} = 5.213$, $p < 0.001$, $d = 0.75$ 95% CI = [−0.43, −0.19]. The ANOVA on the occlusion data yielded a

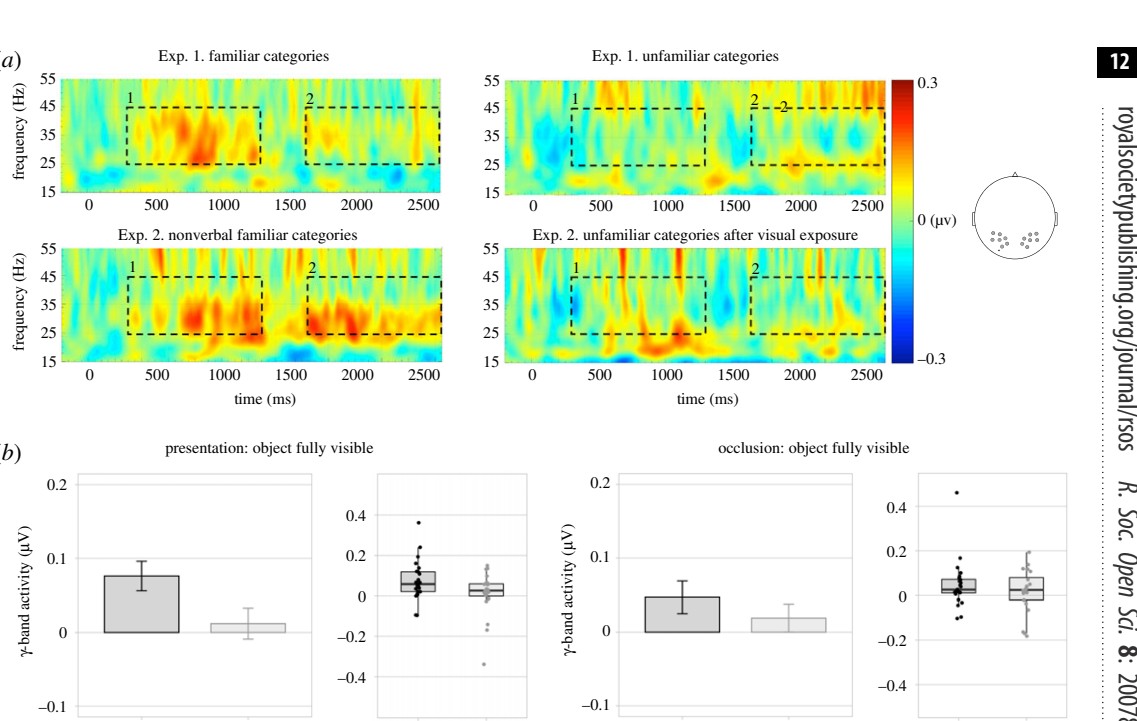

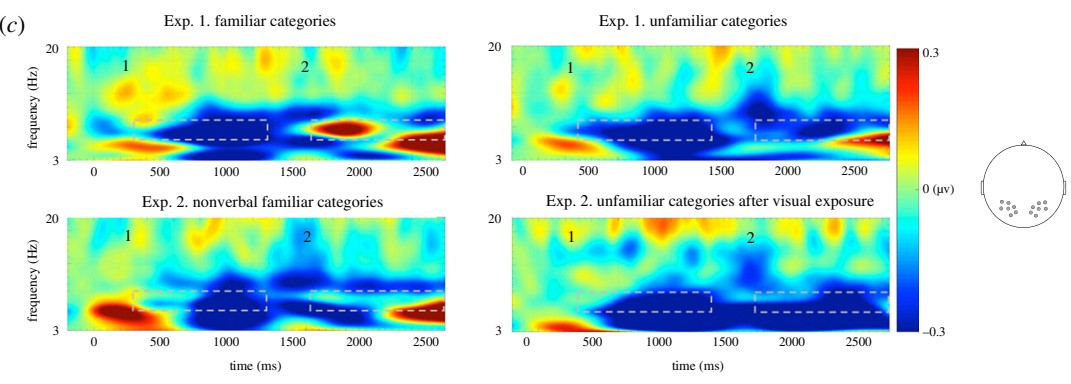

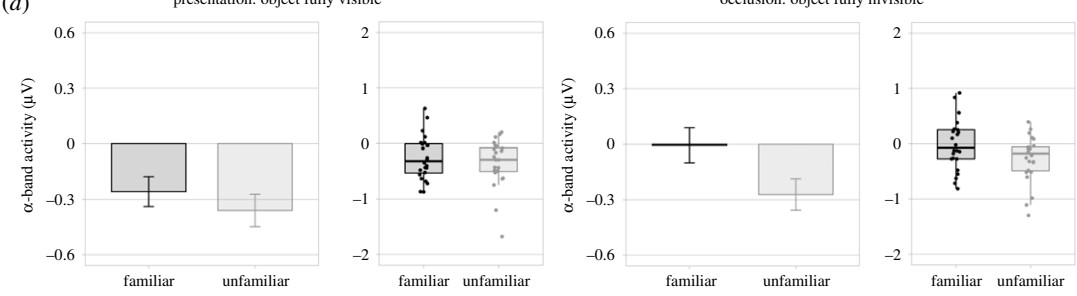

**Figure 3.** Oscillatory activity during object presentation and occlusion in Experiments 1–2: (a,b) γ range (c,d) α range. For visualization purposes, the data were split by experiment. The zero point on the time axis indicates the first frame when the object started to be visible. (a) time–frequency plots represent the changes in the time course of high-frequency oscillations (20–55 Hz) during object presentation (a1) and occlusion (a2), depicting mean baseline-corrected activity averaged across bilateral clusters of electrodes. The dotted areas indicate the time windows (a1: presentation, i.e. object fully visible between 300 and 1300 ms relative to the zero point, a2: occlusion, i.e. object fully invisible between 1650 and 2650 ms relative to the zero point) and the frequency range used for the analysis (γ: 25–45 Hz). (b) Bar plots and boxplots represent mean oscillatory activity in the target frequency range during presentation and occlusion averaged across bilateral clusters of electrodes. Dots represent individual data points. (c) Time–frequency plots represent the changes in the time course of low-frequency oscillations (3–20 Hz) during object presentation (c1) and occlusion (c2), depicting mean baseline-corrected activity averaged across bilateral clusters of electrodes. The dotted areas indicate the time windows and the frequency range used for the analysis (α: 6–9 Hz). (d) Bar plots and boxplots represent mean oscillatory activity in the target frequency range during presentation and occlusion. Dots represent individual data points.

significant main effect of category, $F_{44} = 4.356$, $p = 0.043$, $\eta_p^2 = 0.09$. Because the main effect of hemisphere only approached significance, $F_{44} = 4.016$, $p = 0.051$, $\eta_p^2 = 0.08$, we collapsed the data across hemispheres for further comparisons. A release from suppression was observed during occlusion of the objects from familiar categories: the occlusion α in the familiar-category conditions did not differ from baseline, $t_{23} = 0.065$, $p = 0.945$, 95% CI = [−0.20, 0.19]. By contrast, the activity elicited by the occlusion of objects from unfamiliar categories remained significantly below baseline, $t_{23} = 3.190$, $p = 0.004$, $d = 0.65$, 95% CI = [−0.45, −0.09]. Exploratory analyses performed on the data split by hemisphere yielded the same pattern of results. This suggests that different information maintenance strategies were recruited for familiar versus unfamiliar categories, probably due to distinct representational formats employed by the infants to deal with the storage of categorized versus non-categorized information. On the one hand, the temporal α suppression observed in unfamiliar categories suggests that in the absence of categorization, ventral pathways were involved in the maintenance of sensory information. On the other hand, the release from suppression observed in familiar categories might indicate top-down inhibitory control over visual processing [70]. These distinct patterns of α-band activity recorded in responses to familiar and unfamiliar categories during occlusion are in line with findings that inferior temporal cortices are involved in the sensory processing of visual stimuli, while categorical information is processed in the pre-frontal cortex [46,47]. They also constitute preliminary evidence for the idea that categorization might lead infants to use proto-symbolic representations centred around category symbols [33,41] and devoid of episodic information [8,11].

# 5. General discussion

Numerous convergent findings indicate that categorical information is privileged in object processing and representation throughout the lifespan. One fundamental question regarding these categorical biases is whether they derive from lexical symbols used to communicate about objects and categories, or whether they are a product of nonverbal category structures that precede lexical-semantic knowledge in phylogeny and ontogeny. Our results provide empirical support for the latter view. Across two experiments using an EEG change-detection task in preverbal human infants, we observed a striking dissociation between the way 12-month-olds processed objects from familiar and unfamiliar categories, which cannot be accounted for by their knowledge of category labels.

First, we established that infants' object representations contain a large amount of featural detail unless filtered through category knowledge. Twelve-month-olds reliably detected when an object was replaced by another object from a different category following a brief occlusion, irrespective of whether these objects represented familiar or unfamiliar categories. This was evidenced by their Nc responses to object reappearance, with a larger Nc amplitude recorded to across-category object changes than to the reappearance of the object that was initially occluded. By contrast, when an object was replaced by a different object from the same category, only infants presented with unfamiliar categories displayed sensitivity to the change. That is, infants who viewed unfamiliar categories showed a more negative Nc to within-category object changes than to no change events, unlike infants who viewed familiar categories and whose Nc did not differentiate between these conditions. This asymmetry in within-category change detection confirms that category knowledge influenced the information included by the infants in their object representations. More specifically, category knowledge interfered with the infants' ability to represent individual-specific detail, otherwise readily encoded in its absence. Diverging patterns of oscillatory activity elicited by familiar and unfamiliar categories before and during occlusion suggest that infants rapidly categorized the familiar objects, and categorical information become the core content of the representations they set up.

Second, we demonstrated that the recorded representational changes were triggered by nonverbal category knowledge and independent from language. More specifically, in Experiment 1, we showed that the bias to disregard individual featural details occurred naturally when children observed categories learned outside the laboratory. In Experiment 2, we established that this bias could be rapidly induced by learning new nonverbal categories. This finding confirmed that the recorded categorical modulation of object representation was not reliant on language. Moreover, we found no differences in the patterns of the ERP or oscillatory responses between potentially lexicalized categories acquired in real-life circumstances and nonverbal categories trained in the laboratory. This suggests that, in both experiments, infants probably relied on nonverbal category representations.

Note that we ruled out low-level perceptual explanations for the infants' responses. We used the same task and the same unfamiliar stimuli across three conditions (unfamiliar categories in Experiment 1,

category training in Experiment 2 and control in Experiment 2) and manipulated only the infants' ability to categorize them. Hence, our findings indicate that within-category object changes became challenging to note as a result of learning the relevant category knowledge, and not due to differences in stimulus characteristics or exposure times.

What is the nature of the representations that infants set up to track objects? Do they recruit the same representational formats for objects from familiar versus unfamiliar categories? In principle, mid-level visual representations such as object files [71,72] could be employed in both cases, with bundles of different features bound to them depending on whether the object at hand comes from a familiar or unfamiliar category. Category-diagnostic features could be encoded for familiar categories, while the selection of features would be less constrained for unfamiliar categories, resulting in a random sampling of category-relevant and irrelevant information. Alternatively, however, category knowledge might lead to a qualitative shift towards symbolic representation. By using non-lexicalized categories, we excluded the possibility that infants relied on labels as object placeholders, but they might be using nonverbal symbols indexing familiar categories [33,41]. This model is supported by the patterns of oscillatory α activity recorded during occlusion of the probe objects, indicating a differential response to objects from familiar versus unfamiliar categories. More specifically, temporal α suppression sustained during occlusion of unfamiliar but not familiar objects suggests that sensory featural information was selectively maintained only in the absence of category knowledge. The results of exploratory analyses of γ synchrony also support this idea and call for future research to further corroborate it: the increased activation during presentation of familiar objects might index a rapid recruitment of conceptual symbolic representations carried out upon visual inspection of the familiar objects.

Our findings directly contribute to the debate on how visual working memory develops. Previous work indicates that by six months of age, infants readily represent up to two hidden objects [73] and store certain categorical information characterizing them (e.g. object versus agent: [8]). On the contrary, their ability to remember surface features lags behind [52,74,75], gradually improving until 1 year of age [14,76] and perhaps beyond. The past work has also shown that the storage capacity of infants' working memory is modulated by various factors (e.g. the number of objects to remember, [76]; the time elapsed from hiding, e.g. [74,77]). In particular, Kibbe and Leslie [8] have recently demonstrated that conceptual knowledge increases the number of objects that six-month-olds can remember ([8]; see also, [12,78]). Our findings confirm that older infants can spontaneously encode objects in great detail and this encoding is influenced by category knowledge. Importantly, however, unlike the previous work, we show that categorization appears to decrease the amount of information stored. We believe that although these observations might seem contradictory, they stem from the same underlying process and can provide a further insight into the role that category knowledge might play in working memory. Namely, category knowledge does not simply increase the storage capacity but provides a summary representational format (e.g. category tag or symbol devoid of individual featural information in the same manner as a category label, see also [41]) that allows for information compression and is used by infants to encode and maintain the information. Arguably, this format facilitates setting up and maintaining distinct representations of objects from different categories that fall under different category tags (e.g. baby versus ball, [8]) and may lead to a higher number of such representations to be maintained in memory. At the same time, it appears to compromise the discriminability between individual items from the same category that all fall under the same category tag (e.g. ball 1 versus ball 2, Experiment 1; see also [11]).

Another interesting question is whether categorical object representation that hinders featural encoding arises naturally whenever infants visually inspect an object or has been induced by the nature of our task that involved storing object representations in working memory. Recent research suggests that infants do not spontaneously think of objects in terms of their categories but do so only when required by the nature of the task they are involved in [41]. For example, 12-month-olds have been shown to represent familiar objects under relevant category descriptions when interpreting nonverbal communication while at the same time failing to do so in a non-communicative context. It was argued that this was because interpreting communication triggers the addressee to represent the referent under a conceptual description that is required to make inferences about meaning. It is a possibility that infants in the current task resorted to category-based representations because this representational format is triggered by the demand to maintain information in the working memory.

Functionally, prioritizing category- over featural information might facilitate the acquisition of semantic knowledge, especially needed when linguistic tools of reference disambiguation are unavailable. In particular, before language develops, biases to prioritize category-level representations may ensure that infants map new information directly to categories and not to individuals [33].

The knowledge mapped onto a categorical representation can be directly extended and applied to any individual recognized as an exemplar of the category in question. Under this description, categorical biases in object representation may constitute a mechanism of knowledge generalization ready for use before children come to understand generic language. Such a mechanism could even be helpful in word learning. That is, a word learner biased to build categorical representations of encountered objects would automatically link a novel word (e.g. a common noun 'apple') encountered in the presence of a specific object to its categorical properties, which, in turn, would enable her to accurately extend it to other objects falling in the same category (e.g. the word 'apple' would automatically apply not only to the particular apple that was named but to all members of the same non-linguistic category). Recent research suggests that 12-month-olds might indeed rely on such strategies of word generalization ([31], see also [32] for older infants).

We acknowledge that our results do not rule out the possibility that further modulations of object representation may stem from experience with language. Although specific concepts are differently instantiated in each individual mind [79] (e.g. one's concept of dog will probably be different from that of a veterinary surgeon), we all employ the same category labels in communication, whether to indicate kind membership (e.g. 'this is a dog'), convey generic information (e.g. 'dogs chase cats') or transient individual-specific information (e.g. 'my dog is playing in the yard'). Thus, the use of linguistic symbols that call for category-level representation may further influence the representational structures recruited by object cognition and their contents by virtue of reinforcing the neural connections between labels and category-diagnostic features. In line with this idea, it has been shown that access to category names enhances visual discrimination in match-to-sample tasks [27] and complex visual search contexts [3,80]. On the other hand, natural languages also have tools for communicating about particular individuals (e.g. proper names: Marie Curie, Paris), which have been shown to increase attention to individual-specific features [81,82]. It, thus, seems that some linguistic devices (such as proper names) may further increase sensitivity to the distinctiveness of visual stimuli, while others (such as category labels) decrease it. Whether the latter is simply due to improved categorization performance that enables pre-existing categorical biases to operate or involves independent language-specific computational mechanisms remains to be determined by future research.

To conclude, already in infancy, object representation is not merely a direct function of the features available in perception but is determined by the available category knowledge. Critically, the availability of nonverbal visual categories is sufficient to modulate the contents of object representations by preventing the storage of individual surface features. Hence, at least some forms of categorical biases in object cognition are independent of natural language. Nonverbal symbolic representations might be the basis of generic knowledge transmission and serve as a foundation on which further linguistic modulation of object representation develops.

Ethics. This research was approved by the Ethics Committee of Birkbeck College (approval no.: 111236), University of London, London, UK, and conducted according to the principles defined in the Declaration of Helsinki. Prior to their participation in the research, all families provided written informed consent.

Data accessibility. The data analysed in this manuscript and the code used for data analysis are available in the following OSF repository: https://osf.io/b36cg/ [83].

Authors' contributions. B.P. and T.G. designed research. B.P. collected the data. B.P. and T.G. analysed the data and wrote the manuscript.

Competing interests. We declare we have no competing interests.

Funding. This research was supported by the European Commission Marie Curie Initial Training Networks (grant no. 264301) and the UK Medical Research Council (grant no. G0701484).

Acknowledgements. We thank Jennifer Cruz for help with data acquisition and stimuli preparation; Gergely Csibra, Ansgar Endress, Carina de Klerk, Johannes Mahr, Eugenio Parise, Barbu Revencu and Denis Tatone for comments on a previous version of this manuscript; Natasha Kirkham for feedback on the early task design; Anna Zamm for introduction to EEGLAB.

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
