## [Peer Review File · Royal Society Open Science]

Review History

RSOS-200782.R0 (Original submission)

Review form: Reviewer 1

Is the manuscript scientifically sound in its present form?

Yes

Are the interpretations and conclusions justified by the results?

Yes

Is the language acceptable?

Yes

Do you have any ethical concerns with this paper?

No

Have you any concerns about statistical analyses in this paper?

Yes

Recommendation?

Accept with minor revision (please list in comments)

Comments to the Author(s)

Thank you for the opportunity to read and review this interesting manuscript. The authors test the hypothesis that category knowledge influences infants' object representations. The authors acknowledge that others have investigated this, but they note that it is difficult to rule out the potential role of language in these previous studies since they used nameable objects. The authors therefore used a training study to examine category knowledge without lexical knowledge.

The methods are sound and the results are, for the most part, clear. The results add to our growing understanding of the role of category information in infants' object representations. I have several comments aimed at improving the manuscript.

1) Theoretical background

There are a few places where the wording in the manuscript mischaracterize some of the previous work cited and/or the novelty of the contribution of the current work.

- In the Introduction, the authors state that "infants draw on category but not featural information when individuating objects and storing them in working memory". Infants (and adults for the matter) can and do use features to individuate and store representations of objects (see, for example, Teresa Wilcox's extensive body of work on individuation-by-feature, or other work by Kibbe & Leslie cited by the authors in the discussion).

- In the General Discussion (bottom of page 21 to top of page 23), the authors suggest that their work shows that working memory development does not proceed as a linear increase in capacity, but that the information stored in WM is modulated by category knowledge. However, the idea that what is stored in object representations is modulated by a variety of factors - and therefore development of WM capacity does not proceed linearly - has been argued in many of the papers the authors cite here (for example, Kibbe & Leslie 2013 show that the amount of featural detail infants encode depends on the number of objects hidden; Kibbe & Leslie 2019 show that infants remember more categorical identities than featural identities).

2) Methodological issues

- The authors state that they removed sequences in which infants "blinked". It is not clear what "sequence" refers to here. If sequence is a trial, that is a long time to expect an infant not to blink. If it is a shorter period of time, it is unclear how coding of blinks was synchronized to these shorter windows. More information is needed here.

- The number of artifact-free trials are reported, but it is not clear how many trials infants saw total. Were there differences in the total number of trials between conditions?

3) Analyses

- In analyzing activation in the gamma range, the authors interpret non-significant results as significant ("significant at the tendency level"). The authors specified a priori in their power analyses that alpha was set to .05, and therefore cannot interpret a p value above .05 as significant. This makes any follow-up analyses unjustified by the outcome of the omnibus ANOVA. The authors can still include these additional analyses (separate ANOVAs within each phase) but need to flag that these are post hoc and exploratory, and therefore must be interpreted with extreme caution. This also means that they will need to temper their arguments about the conclusions that can be drawn about the nature of infants' object representations based on gamma asynchrony during presentation in the GD (p. 20, 2nd para). This should be flagged as needing to be interpreted with caution or not included at all.

- The authors again interpret a non-significant effect of hemisphere in the occlusion phase on alpha activation (p. 18). Again, this result needs to be described as non-significant and interpreted accordingly.

Review form: Reviewer 2

Is the manuscript scientifically sound in its present form?

Yes

Are the interpretations and conclusions justified by the results?

Yes

Is the language acceptable?

Yes

Do you have any ethical concerns with this paper?

No

Have you any concerns about statistical analyses in this paper?

No

Recommendation?

Accept with minor revision (please list in comments)

Comments to the Author(s)

This manuscript describes a study examining categorization based on nonverbal information in 12-month-old infants. Two different approaches to EEG analyses were carried out on the data: ERP analyses of the Nc component associated with infant visual attention, and analyses of alpha and gamma oscillations during object occlusion. I believe the paper is relatively strong and has potential to contribute to scientific knowledge regarding the development of categorization. I do have a few specific concerns and suggestions included below.

1. On page 6 line 19, the Coch and Gullick (2012) is really not an appropriate single study to cite when introducing the Nc component. The Nc component has been systematically studied for decades, going back to the early work in the 1980s by Courchesne and colleagues, and citing one review chapter does not appropriately acknowledge the foundational studies that were involved in identifying the Nc and determining the processes Nc is associated with - especially since the authors of the review chapter were not involved in any studies directly focused on Nc.
2. On page 6 line 21, similar to point 1, no studies have definitively shown that larger amplitudes of Nc are diagnostic of larger sensitivity. In general, Nc has been shown to reflect attentional engagement, not degree of perceptual sensitivity.
3. On page 10, can the authors address potential concerns from readers that the high-pass filter setting (.3 Hz) is a bit high and may have introduced artifact?
4. On page 10, the electrode cluster used to examine Nc was quite large (29 electrodes) and covers a pretty broad area of the scalp. Can the authors include a bit more on what led to their decision to use such a large cluster?

5. On page 12, second paragraph under Stimuli, Design, and Procedure, please include more detail on the category-training procedure. For example, how many exemplars were provided per category? Was the actor talking? How long were the videos? What was done if the infant failed to remain attentive to the video during training?

6. First sentence at the top of page 16, these two inferences are pretty significant, and some may consider each to be a bit of a stretch. Citing only a single study to support each inference seems insufficient.

Decision letter (RSOS-200782.R0)

Dear Dr Pomiechowska

On behalf of the Editors, we are pleased to inform you that your Manuscript RSOS-200782 "Nonverbal category knowledge limits the amount of information encoded in object representations: EEG evidence from 12-month-old infants" has been accepted for publication in Royal Society Open Science subject to minor revision in accordance with the referees' reports. Please find the referees' comments along with any feedback from the Editors below my signature.

Please submit your revised manuscript and required files (see below) no later than 7 days from today's (ie 12-Jan-2021) date. Note: the ScholarOne system will 'lock' if submission of the revision is attempted 7 or more days after the deadline. If you do not think you will be able to meet this deadline please contact the editorial office immediately.

on behalf of Dr Emma Hayiou-Thomas (Associate Editor) and Essi Viding (Subject Editor)
openscience@royalsociety.org

Associate Editor Comments to Author (Dr Emma Hayiou-Thomas):

Associate Editor: 1

Comments to the Author:

Thank you for this interesting and well-written paper - I enjoyed reading it, and agree with both reviewers that it has the potential to make a useful contribution to the literature. The reviewers have made a number of thoughtful and constructive suggestions for further improving the manuscript, both in terms of some of the descriptions of prior empirical (R1) and methodological (R2) literature, as well as in the interpretation of the analyses of the gamma and alpha oscillations (R1). Please make sure to address each of their points in your revision of the manuscript.

Reviewer comments to Author:

Reviewer: 1

Comments to the Author(s)

Thank you for the opportunity to read and review this interesting manuscript. The authors test the hypothesis that category knowledge influences infants' object representations. The authors acknowledge that others have investigated this, but they note that it is difficult to rule out the potential role of language in these previous studies since they used nameable objects. The authors therefore used a training study to examine category knowledge without lexical knowledge.

The methods are sound and the results are, for the most part, clear. The results add to our growing understanding of the role of category information in infants' object representations. I have several comments aimed at improving the manuscript.

1) Theoretical background

There are a few places where the wording in the manuscript mischaracterize some of the previous work cited and/or the novelty of the contribution of the current work.

- In the Introduction, the authors state that "infants draw on category but not featural information when individuating objects and storing them in working memory". Infants (and adults for the matter) can and do use features to individuate and store representations of objects (see, for example, Teresa Wilcox's extensive body of work on individuation-by-feature, or other work by Kibbe & Leslie cited by the authors in the discussion).

- In the General Discussion (bottom of page 21 to top of page 23), the authors suggest that their work shows that working memory development does not proceed as a linear increase in capacity, but that the information stored in WM is modulated by category knowledge. However, the idea that what is stored in object representations is modulated by a variety of factors - and therefore development of WM capacity does not proceed linearly - has been argued in many of the papers the authors cite here (for example, Kibbe & Leslie 2013 show that the amount of featural detail infants encode depends on the number of objects hidden; Kibbe & Leslie 2019 show that infants remember more categorical identities than featural identities).

2) Methodological issues

- The authors state that they removed sequences in which infants "blinked". It is not clear what "sequence" refers to here. If sequence is a trial, that is a long time to expect an infant not to blink. If it is a shorter period of time, it is unclear how coding of blinks was synchronized to these shorter windows. More information is needed here.

- The number of artifact-free trials are reported, but it is not clear how many trials infants saw total. Were there differences in the total number of trials between conditions?

3) Analyses

- In analyzing activation in the gamma range, the authors interpret non-significant results as significant ("significant at the tendency level"). The authors specified a priori in their power analyses that alpha was set to .05, and therefore cannot interpret a p value above .05 as significant. This makes any follow-up analyses unjustified by the outcome of the omnibus ANOVA. The authors can still include these additional analyses (separate ANOVAs within each phase) but need to flag that these are post hoc and exploratory, and therefore must be interpreted with extreme caution. This also means that they will need to temper their arguments about the conclusions that can be drawn about the nature of infants' object representations based on gamma asynchrony during presentation in the GD (p. 20, 2nd para). This should be flagged as needing to be interpreted with caution or not included at all.

- The authors again interpret a non-significant effect of hemisphere in the occlusion phase on alpha activation (p. 18). Again, this result needs to be described as non-significant and interpreted accordingly.

Reviewer: 2

Comments to the Author(s)

This manuscript describes a study examining categorization based on nonverbal information in 12-month-old infants. Two different approaches to EEG analyses were carried out on the data: ERP analyses of the Nc component associated with infant visual attention, and analyses of alpha and gamma oscillations during object occlusion. I believe the paper is relatively strong and has potential to contribute to scientific knowledge regarding the development of categorization. I do have a few specific concerns and suggestions included below.

1. On page 6 line 19, the Coch and Gullick (2012) is really not an appropriate single study to cite when introducing the Nc component. The Nc component has been systematically studied for decades, going back to the early work in the 1980s by Courchesne and colleagues, and citing one review chapter does not appropriately acknowledge the foundational studies that were involved in identifying the Nc and determining the processes Nc is associated with - especially since the authors of the review chapter were not involved in any studies directly focused on Nc.
2. On page 6 line 21, similar to point 1, no studies have definitively shown that larger amplitudes of Nc are diagnostic of larger sensitivity. In general, Nc has been shown to reflect attentional engagement, not degree of perceptual sensitivity.
3. On page 10, can the authors address potential concerns from readers that the high-pass filter setting (.3 Hz) is a bit high and may have introduced artifact?
4. On page 10, the electrode cluster used to examine Nc was quite large (29 electrodes) and covers a pretty broad area of the scalp. Can the authors include a bit more on what led to their decision to use such a large cluster?
5. On page 12, second paragraph under Stimuli, Design, and Procedure, please include more detail on the category-training procedure. For example, how many exemplars were provided per category? Was the actor talking? How long were the videos? What was done if the infant failed to remain attentive to the video during training?
6. First sentence at the top of page 16, these two inferences are pretty significant, and some may consider each to be a bit of a stretch. Citing only a single study to support each inference seems insufficient.

===PREPARING YOUR MANUSCRIPT===

===PREPARING YOUR REVISION IN SCHOLARONE===

- An individual file of each figure (EPS or print-quality PDF preferred [either format should be produced directly from original creation package], or original software format).
 - An editable file of each table (.doc, .docx, .xls, .xlsx, or .csv).
 - An editable file of all figure and table captions.
- Note: you may upload the figure, table, and caption files in a single Zip folder.
- Any electronic supplementary material (ESM).
 - If you are requesting a discretionary waiver for the article processing charge, the waiver form must be included at this step.
 - If you are providing image files for potential cover images, please upload these at this step, and inform the editorial office you have done so. You must hold the copyright to any image provided.
 - A copy of your point-by-point response to referees and Editors. This will expedite the preparation of your proof.

- Ensure that your data access statement meets the requirements at <https://royalsociety.org/journals/authors/author-guidelines/#data>. You should ensure that you cite the dataset in your reference list. If you have deposited data etc in the Dryad repository, please only include the 'For publication' link at this stage. You should remove the 'For review' link.
- If you are requesting an article processing charge waiver, you must select the relevant waiver option (if requesting a discretionary waiver, the form should have been uploaded at Step 3 'File upload' above).
- If you have uploaded ESM files, please ensure you follow the guidance at <https://royalsociety.org/journals/authors/author-guidelines/#supplementary-material> to include a suitable title and informative caption. An example of appropriate titling and captioning may be found at https://figshare.com/articles/Table_S2_from_Is_there_a_trade-off_between_peak_performance_and_performance_breadth_across_temperatures_for_aerobic_scope_in_teleost_fishes_/3843624.

Author's Response to Decision Letter for (RSOS-200782.R0)

See Appendix A.

Decision letter (RSOS-200782.R1)

Dear Dr Pomiechowska,

It is a pleasure to accept your manuscript entitled "Nonverbal category knowledge limits the amount of information encoded in object representations: EEG evidence from 12-month-old infants" in its current form for publication in Royal Society Open Science.

You can expect to receive a proof of your article in the near future. Please contact the editorial office (openscience@royalsociety.org) and the production office (openscience_proofs@royalsociety.org) to let us know if you are likely to be away from e-mail contact – if you are going to be away, please nominate a co-author (if available) to manage the proofing process, and ensure they are copied into your email to the journal.

on behalf of Dr Emma Hayiou-Thomas (Associate Editor) and Essi Viding (Subject Editor)
openscience@royalsociety.org

Appendix A

Associate Editor Comments to Author (Dr Emma Hayiou-Thomas):

Associate Editor: 1

Comments to the Author:

Thank you for this interesting and well-written paper - I enjoyed reading it, and agree with both reviewers that it has the potential to make a useful contribution to the literature. The reviewers have made a number of thoughtful and constructive suggestions for further improving the manuscript, both in terms of some of the descriptions of prior empirical (R1) and methodological (R2) literature, as well as in the interpretation of the analyses of the gamma and alpha oscillations (R1). Please make sure to address each of their points in your revision of the manuscript.

Thank you for your letter.

We are delighted that you and the reviewers found our paper "Nonverbal category knowledge limits the amount of information encoded in object representations: EEG evidence from 12-month-old infants" interesting and we were given the opportunity to revise it.

We have very much appreciated the reviewer's constructive comments, which helped us to improve the manuscript. In our revision, we attended to all the issues raised and we attach a detailed response. In addition, we modified our Figure 1 to better represent our experimental design.

We hope that you find our revisions satisfactory and consider our manuscript suitable for publication.

Reviewer comments to Author:

Reviewer: 1

Comments to the Author(s)

Thank you for the opportunity to read and review this interesting manuscript. The authors test the hypothesis that category knowledge influences infants' object representations. The authors acknowledge that others have investigated this, but they note that it is difficult to rule out the potential role of language in these previous studies since they used nameable objects. The authors therefore used a training study to examine category knowledge without lexical knowledge.

The methods are sound and the results are, for the most part, clear. The results add to our growing understanding of the role of category information in infants' object representations. I have several comments aimed at improving the manuscript.

1) Theoretical background

There are a few places where the wording in the manuscript mischaracterize some of the previous work cited and/or the novelty of the contribution of the current work.

- In the Introduction, the authors state that "infants draw on category but not featural information when individuating objects and storing them in working memory". Infants (and adults for the matter) can and do use features to individuate and store representations of objects (see, for example, Teresa Wilcox's extensive body of work on individuation-by-feature, or other work by Kibbe & Leslie cited by the authors in the discussion).

Thank you for bringing our attention to this passage. Indeed, the highlighted sentence does not make justice to the previous infant findings. We did not mean to suggest that

infants are incapable of encoding and using visual features, but only to highlight that they prioritize categorical over featural information, as indicated by a rich number of convergent findings (Kibbe & Leslie, 2019; Bonatti et al., 2002; Surian & Caldi, 2010; Xu, Carey, & Quint, 2004; Feigenson & Halberda, 2008; Stavans et al., 2019). We have now corrected the problematic sentence to clarify this message:

“Like adults, infants privilege category over featural information when individuating objects and storing them in working memory (Kibbe & Leslie, 2019; Bonatti et al., 2002; Surian & Caldi, 2010; Xu, Carey, & Quint, 2004; Feigenson & Halberda, 2008; Stavans et al., 2019) even though they are capable of encoding visual features such as shape, size, pattern and color (Wilcox, 1999).”

p. 3

- In the General Discussion (bottom of page 21 to top of page 23), the authors suggest that their work shows that working memory development does not proceed as a linear increase in capacity, but that the information stored in WM is modulated by category knowledge. However, the idea that what is stored in object representations is modulated by a variety of factors - and therefore development of WM capacity does not proceed linearly - has been argued in many of the papers the authors cite here (for example, Kibbe & Leslie 2013 show that the amount of featural detail infants encode depends on the number of objects hidden; Kibbe & Leslie 2019 show that infants remember more categorical identities than featural identities).

Thank you for pointing out this shortcoming of our general discussion. We have now explicitly acknowledged previous work on different factors modulating the working memory capacity in young infants. We also took care to better explain the novelty of our findings. Namely, we now emphasize that that our experiments provide additional insight into the role of category knowledge in working memory representations. While the previous work shows that category knowledge increases the number of remembered objects (Kibbe & Leslie, 2019, see also Feigenson & Halberda, 2008 for older infants), the present results indicate that category knowledge compromises the encoding of individual surface features of the memorized objects. We propose that this might be because categorization provides a new abstract representational format (e.g., category tag or symbol, Pomiechowska et al., 2019; see also, Xu, 2007) that infants deploy to encode and maintain objects in working memory and that does not contain individual featural information.

“Our findings directly contribute to the debate on how visual working memory develops. Previous work indicates that by 6 months of age, infants readily represent up to two hidden objects (Wynn, 1992) and store certain categorical information characterizing them (e.g., object v. agent: (Kibbe & Leslie, 2019). On the contrary, their ability to remember surface features lags behind (Kaldy & Leslie, 2005; Kibbe & Leslie, 2011; Southgate et al., 2008), gradually improving until one year of age (Kibbe & Leslie, 2013; Wilcox, 1999) and perhaps beyond. The past work has also shown that the storage capacity of infants’ working memory is modulated by various factors (e.g., the number of objects to remember, Kibbe & Leslie, 2013; the time elapsed from hiding, Káldy & Leslie, 2005; Kibbe & Leslie, 2016). In particular, Kibbe and Leslie have recently demonstrated that conceptual knowledge can increase the number of

objects that 6-month-olds can remember (Kibbe & Leslie, 2019; see also, Feigenson & Halberda, 2008; Rosenberg & Feigenson, 2013). Our findings confirm that older infants can spontaneously encode objects in great detail and this encoding is influenced by category knowledge. Importantly, however, unlike in the previous work, we show that categorization appears to decrease the amount of information stored. We believe that although these observations might seem contradictory, they stem from the same underlying process and can provide a further insight into the role that category knowledge might play in working memory. Namely, category knowledge does not simply increase the storage capacity but provides a summary representational format (e.g., category tag or symbol devoid of individual featural information in the same manner as a category label, see also Pomiechowska et al., in press) that allows for information compression and is used by infants to encode and maintain the information. Arguably, this format facilitates setting up and maintaining distinct representations of objects from different categories that fall under different category tags (e.g., baby v. ball, Kibbe & Leslie, 2019) and may lead to a higher number of such representations to be maintained in memory. At the same time, it appears to compromise the discriminability between individual items from the same category that all fall under the same category tag (e.g., ball 1 v. ball 2, Experiment 1; see also Xu, Carey, & Quint, 2004)."

pp. 23

2) Methodological issues

- The authors state that they removed sequences in which infants "blinked". It is not clear what "sequence" refers to here. If sequence is a trial, that is a long time to expect an infant not to blink. If it is a shorter period of time, it is unclear how coding of blinks was synchronized to these shorter windows. More information is needed here.

Thank you for this important clarification question. We have now expanded the description of data analysis protocol to disambiguate this issue.

pp. 9-10

We summarise our amendments below.

First, our continuous EEG contained event flags for (1) object appearance before occlusion, (2) object reappearance after occlusion that we later used for epoching. For each infant, her EEG recording was coupled with a time-locked video recording of the session. Based on the video inspection, we marked as invalid, hence excluded from further analysis, all events during which infants looked away from the screen. Additionally, if on a given trial infants did not attend to the screen before occlusion, we also excluded the following object reappearance event. This is because lack of exposure to the object before occlusion renders impossible change detection after occlusion, a phenomenon targeted by our ERP analysis.

Second, the data were inspected for blinks after segmentation using an automated artefact detection procedure. All segments contaminated by blinks were rejected.

We would also like to note that spontaneous blinking in human infants occurs at a significantly lower frequency than in human adults (i.e., on average at one year of age infants have been observed to make 2 blinks per minute, while adults blink on average 15 times per minute, Zametkin et al., 1979; for a review, Bacher & Smotherman, 2004). Therefore, contamination by blinks is lower in the infant than adult data.

- The number of artifact-free trials are reported, but it is not clear how many trials infants saw total. Were there differences in the total number of trials between conditions?

Let us clarify that in the manuscript we do not report the numbers of artifact free trials but of artifact-free segments split by the analyses conducted (i.e., the Nc ERP analysis, time frequency analysis) as each analysis involved a different segmentation (please see the response above). As we have already reported in the initial version of the manuscript, for both analyses participants contributed comparable numbers of artifact-free trials between conditions, thus ensuring a comparable signal-to-noise ratio between conditions.

We have now reported also the numbers of segments contributed by the infants before rejecting the data contaminated by artefacts and provided the statistical comparisons between conditions. This additional information can be found in the *Data Analysis* sections describing the respective analyses:

ERP analysis, Experiment 1, pp. 10-11

ERP analysis, Experiment 2, p. 15

Time-frequency analysis, pp. 18-19

3) Analyses

- In analyzing activation in the gamma range, the authors interpret non-significant results as significant ("significant at the tendency level"). The authors specified a priori in their power analyses that alpha was set to .05, and therefore cannot interpret a p value above .05 as significant. This makes any follow-up analyses unjustified by the outcome of the omnibus ANOVA. The authors can still include these additional analyses (separate ANOVAs within each phase) but need to flag that these are post hoc and exploratory, and therefore must be interpreted with extreme caution. This also means that they will need to temper their arguments about the conclusions that can be drawn about the nature of infants' object representations based on gamma asynchrony during presentation in the GD (p. 20, 2nd para). This should be flagged as needing to be interpreted with caution or not included at all.

Thank you for pointing out this incongruence in our statistical reporting of the of the time-frequency data. We have now corrected this mistake and indicated that our additional analyses were exploratory. Following the recommendation of the reviewer, we also indicate in the general discussion that interpretation of the gamma-band results should be cautious and further research is needed to validate it.

"The ANOVA in the gamma range (25-45 Hz) revealed two effects that approached significance: an interaction between experiment, category and phase, $F(1,44) = 2.818$, $p = .100$, $\eta p2 = .06$, and a main effect of category knowledge, $F(1, 44) = 2.772$, $p = .106$, $\eta p2 = .06$. Because previous findings suggested that in 12-month-olds gamma-band activity might be an index of conceptual processing during visual inspection of familiar objects

(Gliga, Volein, & Csibra, 2008), we carried out exploratory analyses to probe whether such modulation was manifest also in the present data. Separate ANOVAs with category knowledge and experiment were conducted within each phase (presentation v. occlusion)."

p. 19

- The authors again interpret a non-significant effect of hemisphere in the occlusion phase on alpha activation (p. 18). Again, this result needs to be described as non-significant and interpreted accordingly.

We beg to differ as we did not make any make any claims about the potential hemispheric differences. We simply reported the data split by hemisphere to illustrate that the effects observed on the data collapsed by hemisphere hold when analysed split by hemisphere despite the fact that the main effect of hemisphere approached significance ($p = .051$) in the omnibus ANOVA.

We acknowledge, however, that this way of statistical reporting might have been confusing and have corrected the problematic passages in the revised version of the manuscript. We now **report only the results of the analyses performed on the data collapsed across hemispheres.**

"Because the main effect of hemisphere only approached significance, $F(44) = 4.016$, $p = 0.051$, $\eta_p^2 = 0.08$, we collapsed the data across hemispheres for further comparisons. A release from suppression was observed during occlusion of the objects from familiar categories: the occlusion alpha in the familiar category conditions did not differ from baseline, $t(23) = .065$, $p = .945$, 95% CI = [-.20, .19]. In contrast, the activity elicited by the occlusion of objects from unfamiliar categories remained significantly below baseline, $t(23) = 3.190$, $p = .004$, $d = .65$, 95% CI = [-.45, -.09]. Exploratory analyses performed on the data split by hemisphere yielded the same pattern of results."

p. 20

Reviewer: 2

Comments to the Author(s)

This manuscript describes a study examining categorization based on nonverbal information in 12-month-old infants. Two different approaches to EEG analyses were carried out on the data: ERP analyses of the Nc component associated with infant visual attention, and analyses of alpha and gamma oscillations during object occlusion. I believe the paper is relatively strong and has potential to contribute to scientific knowledge regarding the development of categorization. I do have a few specific concerns and suggestions included below.

1. On page 6 line 19, the Coch and Gullick (2012) is really not an appropriate single study to cite when introducing the Nc component. The Nc component has been systematically studied for decades, going back to the early work in the 1980s by Courchesne and colleagues, and citing one review chapter does not appropriately acknowledge the foundational studies that were involved in identifying the Nc and determining the processes Nc is associated with - especially since the authors of the review chapter were not involved in any studies directly focused on Nc.

We have now extended our reference list beyond the review article and included the foundational studies on the Nc component: Courchesne (1997); Webb, Long, & Nelson (2005); Reynolds & Richards (2005); Richards (2003).

p. 5

2. On page 6 line 21, similar to point 1, no studies have definitively shown that larger amplitudes of Nc are diagnostic of larger sensitivity. In general, Nc has been shown to reflect attentional engagement, not degree of perceptual sensitivity.

Thank you for this comment. It allowed us to clarify our methodological assumptions: in the present design, differences in attentional engagement across conditions could only stem from differences in infants' change detection performance.

We edited the passage highlighted by the reviewer to fully explain our reasoning:

"The Nc is an event-related component observed in the infant EEG at frontocentral sites following modifications in stimulus appearance, reflecting allocation of attention (Courchesne (1997); Webb, Long, & Nelson (2005); Reynolds & Richards (2005); Richards (2003); for a review, Coch & Gullick, 2012). Hence, differences in Nc amplitudes between conditions would indicate differences in the infants' attentional engagement. In our design the only aspect of the stimulus that varied across conditions and could modulate infants' attention was the identity of the object revealed after occlusion. Therefore, we reasoned that the Nc component would index the sensitivity to different object changes (i.e., with larger amplitudes diagnostic of larger sensitivity) and provide a window into the contents of the underlying object representations."

p. 5

3. On page 10, can the authors address potential concerns from readers that the high-pass filter setting (.3 Hz) is a bit high and may have introduced artifact?

We applied 0.3 Hz following the ERP analysis protocol in Parise & Csibra (2012). According to the literature major distortions of the ERP waveforms are caused by low cut offs superior to 0.5 Hz and high cut offs inferior to 10 Hz (Luck, 2014; Rousselet, 2012).

We have now included this information in the manuscript:

"The continuous EEG was offline bandpass filtered at 0.3-100 Hz (following Parise & Csibra, 2012). Although filtering can distort the time course and amplitude of the ERP waveforms, such distortions typically occur when the low cut off exceeds 0.5 Hz or when the high cut off is below 10 Hz (Luck, 2014; Rousselet, 2012)."

p. 10

4. On page 10, the electrode cluster used to examine Nc was quite large (29 electrodes) and covers a pretty broad area of the scalp. Can the authors include a bit more on what led to their decision to use such a large cluster?

The literature indicates that Nc has a broad topography spanning fronto-central sites between Fz and Cz (e.g., Reynolds & Richards, 2005; Richards, 2003). We thus selected a large cluster of electrodes covering both sites.

We now provide this background information in the manuscript:

"As the Nc component is well described in time and space, we determined the specific cluster of electrodes and time window of analysis based on the prior literature showing a broad central to anterior distribution of the Nc (spanning between Cz and Fz, e.g., Reynolds & Richards, 2005; Richards, 2003) and computed the mean Nc amplitude using a cluster of electrodes covering central and anterior sites (3, 4, 5, 6, 7, 10, 11, 12, 13, 16, 18, 19, 20, 23, 24, 28, 29, 30, 35, 36, 104, 105, 106, 110, 111, 112, 117, 118, 124) between 350 and 800 ms following the reappearance of the object (i.e., relative to Time 0 of the segment)."

p. 10

5. On page 12, second paragraph under Stimuli, Design, and Procedure, please include more detail on the category-training procedure. For example, how many exemplars were provided per category? Was the actor talking? How long were the videos? What was done if the infant failed to remain attentive to the video during training?

Thank you for these clarification questions. We have now moved the detailed descriptions of our category-training procedure from the online supplement to the main manuscript:

"During category-training, infants watched videos of an experimenter sorting category exemplars (depicted on picture cards) into two locations based on their category membership (e.g., watering cans placed to the left v. staplers placed to the right, forming two category-based homogenous object sets). In each trial, an actress was presented

behind a table with two shelves, one on the right, containing three exemplars of category A, and one on the left, containing three exemplars of category B. First, the actress greeted the infant ("Hello baby!") while waving and smiling at her. Then, she retrieved from behind the table a picture card representing a new category token, presented it to the infant saying "Look at this!", placed it on one of the two shelves and remained immobile looking in the direction of the placed object (please see Figure 1 for the final display). The actress always sorted the new picture in agreement with the category of the represented object, while the side of placement (left v. right) was counterbalanced. Images of the target items were glued onto black pieces of cardboard and used to make video stimuli. Trials were separated by a short centrally displayed attention getter. There were 4 trials per category, for a total of 8 videos. One trial lasted 15 seconds, for a total duration of the testing session of approximately 2 minutes."

p. 14

"All participants who provided enough artefact-free trials to be included in the final EEG sample watched the entirety of the category training or the matched control procedure administered before the EEG task."

p. 15

6. First sentence at the top of page 16, these two inferences are pretty significant, and some may consider each to be a bit of a stretch. Citing only a single study to support each inference seems insufficient.

Thank you for the feedback. We have taken care to provide further references supporting our claims.

The amended passage now reads:

"Gamma-band oscillations have been previously associated with object processing in human adults (Tallon-Baudry, Kreiter, & Bertrand, 1999), human infants (Csibra et al., 2000; Kaufman, Csibra, & Johnson, 2003; 2005; Gliga, Volein, & Csibra, 2010; Southgate et al., 2008), and non-human animals (Lundqvist et al., 2016). Alpha-band oscillations have been argued to regulate working-memory representations (de Vries, Slagter, & Olivers, 2020; Bonnefond & Jensen, 2012; Riddle et al., 2020)."

p. 17